# Characteristic Volatile Fingerprints of Four Chrysanthemum Teas Determined by HS-GC-IMS

**DOI:** 10.3390/molecules26237113

**Published:** 2021-11-24

**Authors:** Zhiling Wang, Yixin Yuan, Bo Hong, Xin Zhao, Zhaoyu Gu

**Affiliations:** Beijing Key Laboratory of Development and Quality Control of Ornamental Crops, Department of Ornamental Horticulture, China Agricultural University, Beijing 100193, China; bs20193170894@cau.edu.cn (Z.W.); yyxyuanyi@cau.edu.cn (Y.Y.); hongbo1203@cau.edu.cn (B.H.)

**Keywords:** chrysanthemum tea, HS-GC-IMS, volatile compound, PLS-DA

## Abstract

Volatile composition is an important feature that determines flavor, which actively affects the overall evaluation of chrysanthemum tea. In this study, HS-GC-IMS (headspace-gas chromatography-ion mobility spectrometry) was performed to characterize the volatile profiles of different chrysanthemum tea subtypes. Forty-seven volatiles of diverse chemical nature were identified and quantified. Partial least squares discriminant analysis (PLS-DA) revealed that four chrysanthemum teas were distinct from each other based on their volatile compounds. Furthermore, this work provides reference methods for detecting novel volatile organic compounds in chrysanthemum tea plants and products.

## 1. Introduction

Chrysanthemum, which is a plant native to China, is one of the top ten traditional flowers in China. It is well known for its various ornamental, medicinal, and edible properties [1,2,3]. For many years, this flower has been cultivated for consumption as food [4,5,6,7]. Globally, tea is one of the most preferred beverages [8], and in Asia, chrysanthemum tea is popular because of its unique flavor [9]. Volatile composition is an important feature contributing to flavor, and it directly influences the overall assessment of chrysanthemum tea and its preference by consumers [10]. Much research has focused on the qualitative aspects of chrysanthemum tea’s function [6,11,12,13,14,15], whereas its volatility remains generally understudied.

As a flower-based tea, chrysanthemum is well liked because of its health benefits, appealing taste, and unique smell, as determined by volatility [10]. Chrysanthemum tea’s volatility and nutritional content are important factors determining its commercial quality [16,17,18]. The common varieties of commercially available chrysanthemum tea are mainly ‘Hangju’ and chamomile. There have been a large number of studies that have analyzed the nutritional components in chrysanthemum tea, such as flavonoids [19]; chlorogenic acid is also a key nutrient in chrysanthemum tea [20]. Nevertheless, volatility is a noteworthy feature of chrysanthemum tea, and it constitutes another direct sensory feature that attracts certain consumers [21,22].

In chrysanthemum cultivars and their wild relatives, the primary volatile compounds are monoterpenoids and oxygenated monoterpenoids, including camphor, α-pinene, chrysanthenone, safranal, myrcene, eucalyptol, 2,4,5,6,7,7ab-hexahydro-1*H*-indene, verbenone, β-phellandrene, and camphene [23]. The primary components in *Chrysanthemum morifolium* volatile oil are monoterpene and sesquiterpene compounds, including hydrocarbons, esters, and organic acids, and the most abundant volatile component is α-curcumene [24]. In *C. morifolium* ‘Boju’, the main volatiles are eucalyptol, fifolone, chrysanthenone, and *cis*-chrysanthenol acetate, among others [25]. In various types of chamomiles, although the main compounds identified are monoterpenoids, sesquiterpenoids, and esters [26], the composition and content of these volatiles are quite different from those in chrysanthemum.

Chrysanthemum tea’s volatility is crucial to consumer choice, and therefore, the detection of volatile organic compounds (VOCs) with volatile components is significant. Recently, there has been increased use of gas chromatography–ion mobility spectrometry (GC-IMS), a new technology capable of detecting VOCs [27]. Specifically, GC-IMS is a rapid detection technique for VOCs in samples, whereby gas chromatography is combined with ion mobility spectroscopy [28]. This technology is based on the separation of complex compounds by a gas chromatography column, where ionized compounds in a specific electric field move through a fixed distance (drift tube) that requires differing drift times to achieve the separation of distinct compounds [29]. Building on that, HS-GC-IMS is a technique that combines GC-IMS with headspace sampling [30] to quickly detect and analyze the VOCs in certain samples; the technique has a low detection limit but high selectivity [31]. HS-GC-IMS has been applied to analyze volatile compounds in grain [32,33], essential oil [31], Chinese medicine [34], wines [35,36], and tea [37].

Herein, we investigated a new chrysanthemum variety, ‘Xiaokuixiang’, with a fruity taste suitable for tea. The main purpose of this study was to compare and analyze the volatile components and contents of ‘Xiaokuixiang’ vis-à-vis ‘Hangju’, ‘Huangju’, and chamomile tea (Figure 1) by GC-IMS and establish their flavor fingerprints. Partial least squares discriminant analysis (PLS-DA) was performed to compare the similarity in volatility between the four chrysanthemum teas. Our results mainly demonstrate that the four chrysanthemum teas were completely separated based on their different volatile substances. Furthermore, this work demonstrates that the different components and contents of volatile substances led to the different aromas of the four chrysanthemum teas.

## 2. Materials and Methods

### 2.1. Materials

The ‘Hangju’, ‘Huangju’, and chamomile used in the experiment came from supermarkets, while ‘Xiaokuixiang’ was a self-bred variety. The latter was harvested and air-dried during the flowering period, and then oven-dried at 40 °C to ensure it was completely dry and, thus, fit for usability.

### 2.2. HS-GC-IMS

From each of the collected four types of dry chrysanthemum teas, we took 100 g and immersed it in 100 mL of 100 °C drinking water and let the tea brew for 30 min. Then, a 0.22 μm filter (organic phase with a needle filter) was used for collecting the chrysanthemum tea without impurities and placing it into an injection bottle (18 mm precision thread vial, volume of 20 mL, 75.5 mm × 22.5 mm, clear glass, rounded bottom, white marking spot, and CNW LOGO, Borosilicate Type I Class A) to determine the VOCs of chrysanthemum teas.

The GC-IMS analysis was entrusted to the Application Shandong Hanon Scientific Instrument Co., Ltd., Jinan, China. Briefly, approximately 5 mL of each chrysanthemum tea type was placed into a 20 mL headspace bottle and incubated at 90 °C for 15 min at 500 rpm before determination. The injection volume was 200 μL, and the injection temperature was 85 °C.

### 2.3. Statistical Analysis

The instrument analysis software of the GC-IMS equipment was used. The NIST (National Institute of Standards and Technology, an official website of the United States government) and IMS (ion mobility spectroscopy, G.M.S., Dortmund, Germany) databases, which are built into this application software, can be and were used for the qualitative analysis of substances encountered in the four tea types.

Data analyses were performed in MS Excel (pie graph), GraphPad Prism 8.0 (bar chart), and Heatmap Illustrator v1.0 (CUCKOO Workgroup, Wuhan, China). All determinations were performed in triplicate, and the results are expressed as the mean ± standard deviation (SD). One-way ANOVA (*p* < 0.05) was performed.

## 3. Results

### 3.1. HS-GC-IMS Topographic Plots for the Different Chrysanthemum Teas

The four chrysanthemum teas are shown in Figure 1. Their volatile compounds were analyzed by HS-GC-IMS, and the ensuing data are depicted in a 2D topographical visualization. As Figure 2 shows, the components and contents of volatiles clearly vary among the different chrysanthemum teas. ‘Xiaokuixiang’ contained the greatest number of volatile compound types, and by contrast, the volatile species of ‘Hangju’ were the least (Figure 2).

To more clearly compare the differences in the volatile components between the four chrysanthemum teas, the topographic plot of fresh stipes from ‘Hangju’ was selected as the reference, from which the topographic plots of the other tea type samples were deducted (Appendix A). Accordingly, if the volatile compounds were consistent, the background after the deduction was white, whereas red or blue indicated that the substance concentration was higher or lower than the reference value, respectively. Most of the signals in the chrysanthemum tea’s topographic plot from the four tea types had retention times between 200 and 1000 s, with several different signals discernible in ‘Xiaokuixiang’. These results showed that the volatiles of ‘Xiaokuixiang’ had the most components and the highest concentration, while ‘Hangju’ had the least in both respects.

### 3.2. Comparative Analysis of Common Volatile Compounds in Different Chrysanthemum Teas

A total of 150 volatile compounds were identified from the four chrysanthemum teas. These compounds were characterized by comparing their IMS drift time and retention index with those of the real reference compound. A total of 47 typical target compounds from topographic plots were identified through the GC × IMS library (Table 1, Appendix A). As shown in Figure 3, qualitative and quantitative analyses were performed on the 47 identified compounds from the different tea samples (Figure 3). A stronger signal indicated a higher volatile content. These results demonstrated that the compound types and contents in ‘Xiaokuixiang’ surpassed those in the other three tea types (Figure 3B).

We then compared the volatile components of each chrysanthemum tea. This revealed that the total amount of volatile compounds was highest in ‘Huangju’ tea, followed by ‘Xiaokuixiang’, with the lowest content of volatile components in ‘Hangju’ (Figure 4). ‘Xiaokuixiang’ had the highest content of ester, ketone, and alcohols compounds, and ‘Huangju’ had the highest content of terpenoids and aldehyde compounds. Compared with the other three chrysanthemums, ‘Hangju’ had the lowest content of ester, alcohols, terpenoids, and aldehyde compounds. The variety of volatile compounds in ‘Xiaokuixiang’ was relatively uniform. Notably, the content of ketone compounds in chamomile was relatively low, and almost no ester compounds were detected in ‘Huangju’ and ‘Hangju’ (Figure 4).

### 3.3. Comparative Analysis of Unique Volatile Compounds in Different Chrysanthemum Teas

To clarify the specific differentially occurring substances in the comparison samples, all peaks obtained by gas-phase ion mobility spectroscopy were selected for a fingerprint comparison. The results showed that the volatile compounds for the four chrysanthemum teas were quite different. Figure 5 shows the visual information pertaining to the types and the corresponding contents of volatile compounds in the chrysanthemum tea types studied. The greatest variety of volatile compounds was found in ‘Xiaokuixiang’, while there were fewer varieties of volatile compounds in ‘Huangju’ as compared to ‘Xiaokuixiang’. Chamomile exhibited fewer types of volatile compounds than ‘Huangju’, while ‘Hangju’ showed the fewest varieties of volatile compounds (Figure 5 and Appendix A). Additionally, we analyzed the primary volatile components and main compounds in each chrysanthemum tea. We found that, in chamomile, the primary volatile components consisted of ethyl 2-methylbutanoate in esters and 1,8-cineole monomer in alcohols (Figure 6A), and for those in ‘Huangju’, it was 1-menthol monomer and 1,8-cineole dimer in terpenoids (Figure 6B). Likewise, those in ‘Xiaokuixiang’ were ethyl 2-methylbutanoate in esters and 6-methyl-5-hepten-2-one in ketones (Figure 6C), and those in ‘Hangju’ were 6-methyl-5-hepten-2-one in ketones and 1-menthol monomer in terpenoids (Figure 6D). Hence, the volatile components of each chrysanthemum tea were not the same.

Our quantitative analysis of the identified compounds showed that there were high amounts of some of them in chamomile, such as 1,8-cineole monomer, ethyl 2-methylbutanoate, 1,8-cineole dimer, 6-methyl-5-hepten-2-one, and hexanal dimer. Likewise, high amounts of ethyl 2-methylbutanoate, 6-methyl-5-hepten-2-one, hexanal dimer, and 1,8-cineole monomer, as well as acetone, were found in ‘Xiaokuixiang’. In ‘Huangju’, high amounts of 1-menthol monomer, along with 1,8-cineole dimer, 6-methyl-5-hepten-2-one, 1,8-cineole monomer, and hexanal dimer were present, and finally, high amounts of 6-methyl-5-hepten-2-one, 1-menthol monomer, 1,8-cineole monomer, hexanal dimer, and hexanal monomer were present in ‘Hangju’ (Figure 6). We determined that ‘Hangju’, chamomile, ‘Xiaokuixiang’, and ‘Huangju’ contained the highest levels of 6-methyl-5-hepten-2-one, 1,8-cineole monomer, ethyl 2-methylbutanoate, and 1-menthol monomer, respectively (Figure 7 and Appendix A).

### 3.4. PLS-DA Analysis of the Four Chrysanthemum Teas

PLS-DA is a multivariate statistical analysis method with supervised pattern recognition. PLS-DA was performed to separate the four chrysanthemum teas based on the volatile compounds after data normalization. As shown in Figure 7A, the four groups of chrysanthemum teas could be completely separated. The relationship between groups chamomile and ‘Hangju’ was more similar, and the relationship between groups ‘Xiaokuixiang’ and ‘Huangju’ was weaker. Furthermore, ‘Xiaokuixiang’ and ‘Huangju’ were distinct from chamomile and ‘Hangju’, respectively. A score plot displays clear separation between the samples. The samples could be separated into four classes with preferably explained variance (R^2^Y = 0.666, Q^2^ = 0.512). The variable importance in projection (VIP) scores are the estimate of the importance of each variable in the PLS model. Among the important volatile compounds that contributed to the separation were ethyl 2-methylbutanoate, (*E*)-2-hexenal dimer, acetone, ethyl 2-methylpropanoate, 1,8-cineole dimer, heptanoic acid dimer, methyl 2-methylbutanoate dimer, propylacetate dimer, linalool oxide monomer, 1-menthol dimer, 1-menthol monomer, (*E*)-2-hexenal monomer, 2-pentenal (*E*) dimer, and 2-methyl butanol dimer (Figure 7B). Taken together, these results revealed that the volatile fingerprints of different chrysanthemum teas were successfully established through GC-IMS.

## 4. Discussion

### 4.1. Volatile Compounds in Different Chrysanthemum Teas

Chrysanthemum tea is one of the most popular flower teas in China because of its unique flavor and nutritional qualities [1]. Most studies have focused on the nutritional and functional components of chrysanthemum tea [4,6,38], but the volatility is also an important trait that determines chrysanthemum tea’s overall quality and flavor [10]. To fill this knowledge gap, in this study, we relied on GC-IMS to identify the volatile components in different chrysanthemum teas. Furthermore, a cluster analysis based on volatile components was used to distinguish chrysanthemum teas with different flavors. Moreover, ‘Xiaokuixiang’ could be a new type of chrysanthemum tea because of its unique volatile components.

Currently, the main types of chrysanthemum tea on the Chinese market are chamomile, ‘Huangju’, and ‘Hangju’ varieties. We detected and analyzed the VOCs of chrysanthemum teas made using those three types and also ‘Xiaokuixiang’. The main volatile compounds in those chamomile tea infusions were aldehydes (34.87%) and terpenoids (30.92%; Figure 6A). A previous study that also used GC-MS found differing components of main volatile compounds in fresh German versus Roman chamomile teas; the former’s major compounds were sesquiterpenoids and monoterpenoids, while the latter’s were esters [26]. In our study, it was obvious that the components of the main volatile compounds are different between the flower’s head and the brewed tea in chamomile. Our research group bred ‘Xiaokuixiang’, and its exploratory potential is promising. Its levels of ketones, aldehydes, esters, and terpenoids were 23.9%, 21.86%, 21.34%, and 18.73%, respectively; hence, the volatile compounds’ distribution of each major category was relatively uniform (Figure 6C). In stark contrast, for ‘Huangju’, the main volatile compounds present in its tea were terpenoids (52.72%) (Figure 6B). To the best of our knowledge, no studies have investigated the components of the main volatile compounds in ‘Huangju’ chrysanthemum tea. Sesquithujene, nootkatene, β-curcumene, cedrol, and β-bisabolol are the primary compounds in ‘Hangju’ [39]. Compared with the volatile compounds in dried chrysanthemum flowers, the main volatile compounds in the ‘Hangju’ tea infusion were aldehydes (33.88%) and ketones (31.69%) (Figure 6D). The components of the main volatile compounds were also different between the flower’s head and tea infusion in ‘Hangju’, similar to the case in chamomile. Therefore, different chrysanthemum teas harbor uniquely dominant VOCs.

### 4.2. Different Chrysanthemum Teas Differed in Their Flavor

We compared the VOC content and compounds in the four chrysanthemum teas. A total of 150 volatile compounds were identified. Approximately 47 compounds were identified in different samples by the qualitative and quantitative analyses (Table 1). As seen in Figure 2, the VOCs for ‘Xiaokuixiang’ were the most abundant and had the highest concentrations, and the lowest amounts of volatile components were found in ‘Hangju’. The volatile compound present at the highest level was also different among the four tea types: 1,8-cineole in chamomile, ethyl 2-methylbutanoate in ‘Xiaokuixiang’, 1-menthol monomer in ‘Huangju’, and 6-methyl-5-hepten-2-one in ‘Hangju’. However, the most abundant volatile compounds in the product may not be the most important factor that contributes to their flavor, because the comprehensive performance of the composition, type, and content proportion of volatile substances constituted the unique flavor of each tea.

Therefore, in this study, the formation of different flavors of the four chrysanthemum teas did not necessarily result from highest amount of chemical compounds, but rather, may have resulted due to the main contribution of volatile compounds present in very low amounts, or the combination of multiple compounds. For example, in ‘Huangju’ and ‘Hangju’, although the amount of esters was low, esters are the main flavor compounds in many fruits [40,41], and therefore, the effect of esters on flavor cannot be ignored. Thus, the formation of the unique flavor of the four chrysanthemum teas was closely related to the combination and amount of their volatiles and the perception threshold of individual volatile compounds.

In this study, PLS-DA was performed as shown in Figure 7A, and ‘Hangju’, ‘Huangju’, chamomile, and ‘Xiaokuixiang’ could be distinguished from each other. Thus, we determined that GC-IMS may be successfully used to distinguish chrysanthemum tea according to the composition of volatile substances. In summation, we found a new type of chrysanthemum tea with a unique flavor that can enrich the varieties of chrysanthemum tea and expand chrysanthemum’s economic value, while also providing new insights and ideas for the study of edible chrysanthemum plant products.

## 5. Conclusions

In this study, we detected and quantified the variation in volatile flavor components in infusions made with four types of chrysanthemum teas. HS-GC-IMS was used to effectively identify characteristic flavor compounds in the different samples. The volatile components and their amounts differed among the tested chrysanthemum teas. The main volatile components in chamomile were ethyl 2-methylbutanoate in esters and 1,8-cineole monomer in alcohols; in ‘Xiaokuixiang’, they were ethyl 2-methylbutanoate in esters and 6-methyl-5-hepten-2-one in ketones. However, in both ‘Hangju’ and ‘Huangju’, they were 6-methyl-5-hepten-2-one in ketones and 1-menthol monomer in terpenoids. The volatile compounds’ PLS-DA analysis showed that the four chrysanthemum teas could be completely separated.

## Figures and Tables

**Figure 1 molecules-26-07113-f001:**
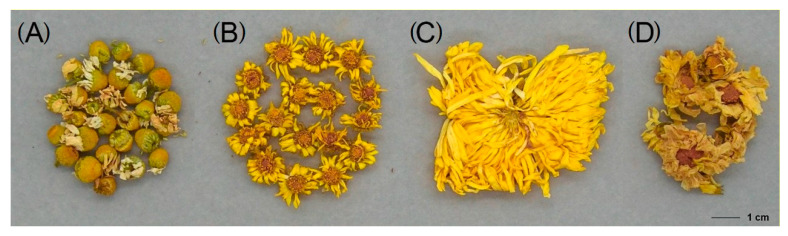
The four types of dry chrysanthemum teas: (**A**) Chamomile; (**B**) ‘Xiaokuixiang’; (**C**) ‘Huangju’; and (**D**) ‘Hangju’.

**Figure 2 molecules-26-07113-f002:**
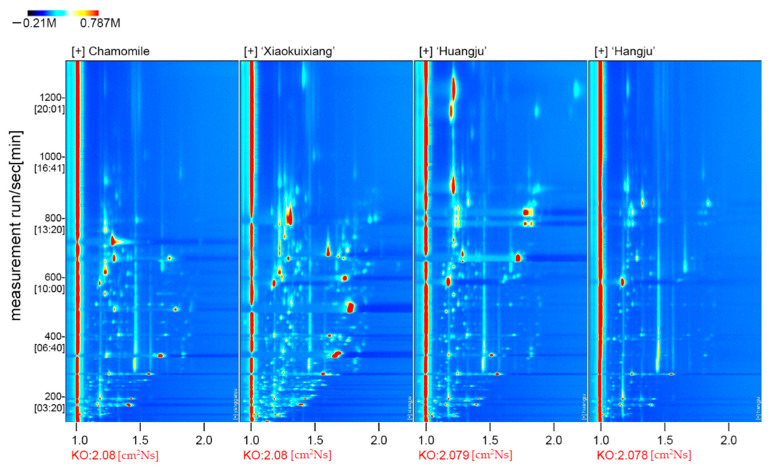
The two-dimensional spectrum of VOCs from the four chrysanthemum teas.

**Figure 3 molecules-26-07113-f003:**
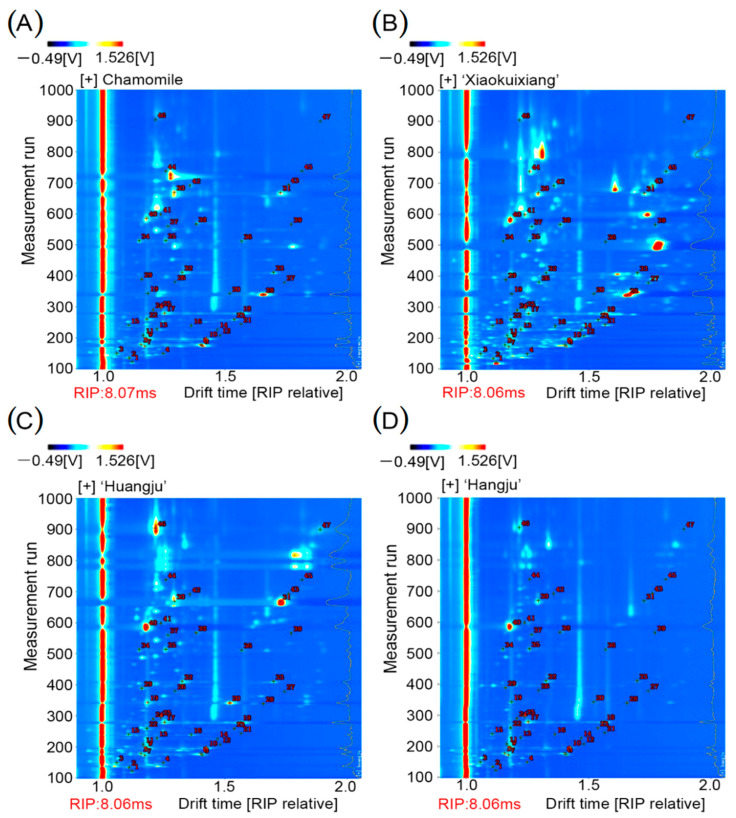
Qualitative results of gas-phase ion mobility spectra for the four chrysanthemum teas: (**A**) Chamomile, (**B**) ‘Xiaokuixiang’, (**C**) ‘Huangju’, (**D**) ‘Hangju’.

**Figure 4 molecules-26-07113-f004:**
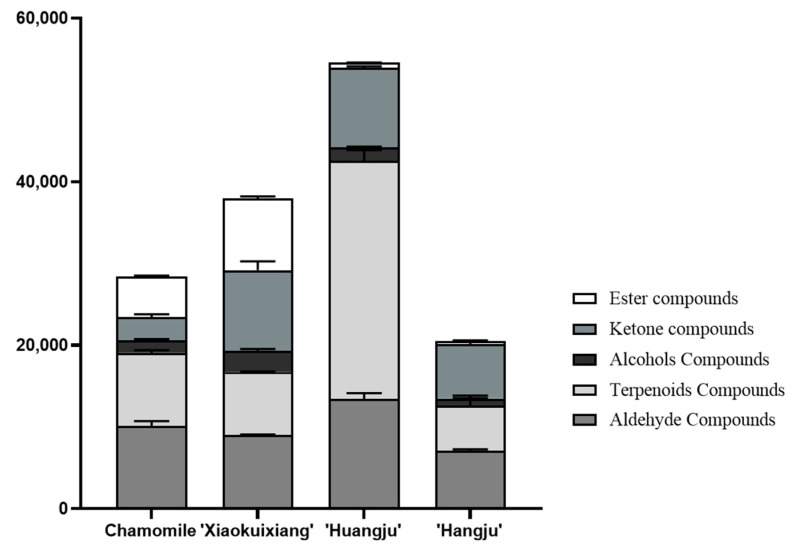
The composition of VOCs in the four chrysanthemum teas. Error bars show the standard deviation between three biological replicates (*n* = 3).

**Figure 5 molecules-26-07113-f005:**
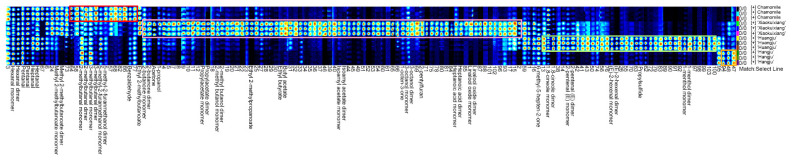
Gallery plot of samples of the four chrysanthemum teas. Each line in the graph represents all the selected signal peaks in a sample. Each column in the diagram represents the signal peak of the same VOC in different samples, and the undetermined substances in the library are replaced by numbers.

**Figure 6 molecules-26-07113-f006:**
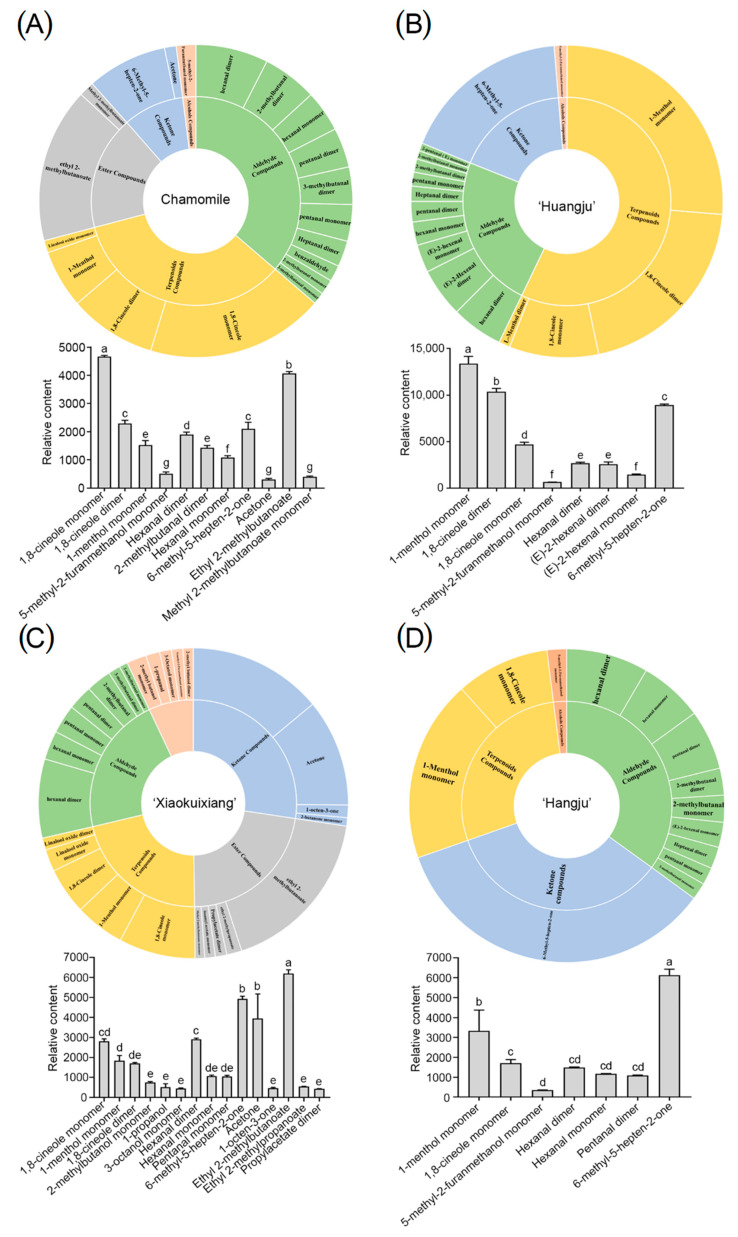
The VOCs in the four chrysanthemum teas: (**A**) The pie chart and column VOCs of Chamomile; (**B**) the pie chart and column VOCs of ‘Huangju’; (**C**) the pie chart and column VOCs of ‘Xiaokuixiang’; (**D**) the pie chart and column VOCs of ‘Hangju’. Pie chart and column, (content > 300), error bars show the standard deviation between three biological replicates (*n* = 3). Three independent experiments were performed and error bars indicate standard deviations. Letters indicate significant differences which were determined by Duncan’s multiple range test (*p* < 0.05).

**Figure 7 molecules-26-07113-f007:**
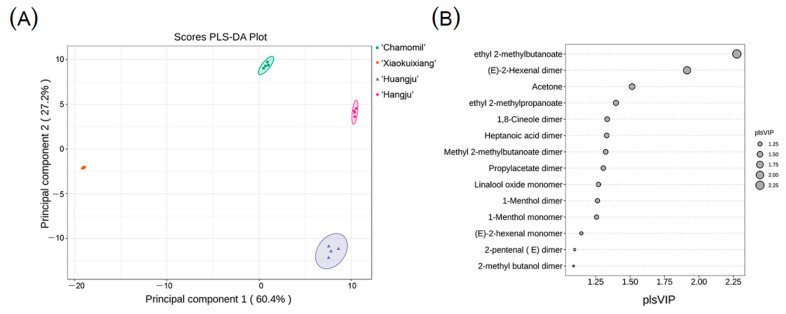
PLS-DA of the four chrysanthemum teas. (**A**) PLS-DA of the four chrysanthemum teas, (**B**) the variable importance in projection (VIP) scores.

**Table 1 molecules-26-07113-t001:** Properties of volatile compounds.

Compound	CAS	Formula	MW	RI	Rt [s]	Dt [a.u.]	Chamomile	‘Xiaokuixiang’	‘Huangju’	‘Hangju’
(*E*)-2-Hexenal dimer	C6728263	C_6_H_10_O	98.1	845	341.96	1.52	119.15 ± 13.38	75.51 ± 3.9	2584.61 ± 200.04	164.85 ± 17.54
(*E*)-2-Hexenal monomer	C6728263	C_6_H_10_O	98.1	845.9	343.05	1.19	266.3 ± 4.4	118.7 ± 13.32	1466.64 ± 58.49	460.29 ± 37.3
1,8-Cineole dimer	C470826	C_10_H_18_O	154.3	1039.4	668.17	1.73	2288.27 ± 95.94	1703.92 ± 33.27	10,356.01 ± 304.58	269.45 ± 17.21
1,8-Cineole monomer	C470826	C_10_H_18_O	154.3	1039.9	669.06	1.30	4666.11 ± 33.93	2808.24 ± 98.28	4684.26 ± 217.57	1709.26 ± 147.07
1-Menthol dimer	C2216515	C_10_H_20_O	156.3	1164.6	899.45	1.89	76.31 ± 8.05	104.58 ± 9.7	611.43 ± 44.5	95 ± 8.35
1-Menthol monomer	C2216515	C_10_H_20_O	156.3	1167.1	904.07	1.22	1523.09 ± 132.67	1832.42 ± 212.92	13,371.96 ± 646.6	3336.42 ± 842.92
1-Octen-3-one	C4312996	C_8_H_14_O	126.2	980.7	559.80	1.27	147.74 ± 10.47	457.74 ± 30.52	185.63 ± 11.63	132.28 ± 8.71
1-Propanol	C71238	C_3_H_8_O	60.1	567.8	133.48	1.11	277.14 ± 37.46	510.58 ± 137.29	285.27 ± 70.16	120.95 ± 13.73
2-Butanone dimer	C78933	C_4_H_8_O	72.1	598.6	148.07	1.25	66.32 ± 8.38	206.39 ± 16.58	91.84 ± 11.82	56.38 ± 2.7
2-Butanone monomer	C78933	C_4_H_8_O	72.1	600.2	148.81	1.06	225.59 ± 5.57	304.15 ± 9.95	256.77 ± 17.37	224.55 ± 3.13
2-Methyl butanol dimer	C137326	C_5_H_12_O	88.1	730.6	226.07	1.47	74.41 ± 43.88	377.13 ± 39.66	61.58 ± 6.67	24.73 ± 1.82
2-Methyl butanol monomer	C137326	C_5_H_12_O	88.1	732.1	227.38	1.22	233.41 ± 45.94	738.55 ± 36.4	173.98 ± 10.38	90.78 ± 9.45
2-Methylbutanal dimer	C96173	C_5_H_10_O	86.1	661.8	177.99	1.40	1432.09 ± 65.61	909.09 ± 49.14	674.59 ± 72.26	495.8 ± 36.75
2-Methylbutanal monomer	C96173	C_5_H_10_O	86.1	659.7	177.00	1.16	401.75 ± 14.95	301.28 ± 4.45	443.43 ± 6.92	492.91 ± 20.22
2-Pentenal (*E*) dimer	C1576870	C_5_H_8_O	84.1	745.3	238.45	1.37	54.63 ± 6.98	120.37 ± 5.89	243.18 ± 31.99	33.63 ± 2.47
2-Pentenal (*E*) monomer	C1576870	C_5_H_8_O	84.1	745.6	238.71	1.11	165.33 ± 11.36	200.09 ± 3.45	426.63 ± 36.76	152.51 ± 7.41
2-Pentylfuran	C3777693	C_9_H_14_O	138.2	1001.9	598.78	1.24	212.31 ± 21.79	1327.04 ± 82.25	263.66 ± 30.66	172.3 ± 2.81
3-Methylbutanal dimer	C590863	C_5_H_10_O	86.1	652.4	173.54	1.41	979.83 ± 46.43	458.85 ± 29.25	219.49 ± 28.19	218.41 ± 19.24
3-Methylbutanal monomer	C590863	C_5_H_10_O	86.1	649.7	172.30	1.17	310.93 ± 22.26	252.22 ± 6.09	251.21 ± 4.01	312.4 ± 14.22
3-Octanol dimer	C589980	C_8_H_18_O	130.2	983.5	565.02	1.77	27.63 ± 0.35	109.59 ± 5.66	32.29 ± 3.77	31.99 ± 6.66
3-Octanol monomer	C589980	C_8_H_18_O	130.2	984.2	566.23	1.39	218.74 ± 9.55	437.37 ± 27.6	232.78 ± 8.53	154.96 ± 5.27
5-Methyl-2-furanmethanol dimer	C3857258	C_6_H_8_O_2_	112.1	954.5	511.57	1.57	197.92 ± 22.36	59.33 ± 1.58	149.97 ± 7.99	102.93 ± 8.99
5-Methyl-2-furanmethanol monomer	C3857258	C_6_H_8_O_2_	112.1	955.8	513.98	1.26	511.85 ± 51.91	411.24 ± 14.62	664.63 ± 13.69	355.18 ± 8.68
6-Methyl-5-hepten-2-one	C110930	C_8_H_14_O	126.2	994.7	585.52	1.18	2100.88 ± 193.06	4922.49 ± 108.53	8939.93 ± 89.23	6135.18 ± 243.3
Acetone	C67641	C_3_H_6_O	58.1	541.7	121.11	1.12	307.59 ± 31.17	3941.64 ± 1000.18	279.57 ± 72.23	93.38 ± 8.36
benzaldehyde	C100527	C_7_H_6_O	106.1	955	512.56	1.15	422.9 ± 56.36	129.43 ± 5.64	251.34 ± 11.67	228.79 ± 1.29
Butyl acetate	C123864	C_6_H_12_O_2_	116.2	804.6	294.95	1.24	21.92 ± 1.42	97.13 ± 4.61	38.21 ± 6.31	21.04 ± 2.4
Ethyl 2-methylbutanoate	C7452791	C_7_H_14_O_2_	130.2	841.7	338.13	1.66	4068.59 ± 54.94	6201.27 ± 149.74	105.39 ± 2.41	45.64 ± 2.85
Ethyl 2-methylpropanoate	C97621	C_6_H_12_O_2_	116.2	749.8	242.27	1.57	45.89 ± 10.29	544.37 ± 4.24	23.53 ± 1.3	23.09 ± 3.23
Ethyl butyrate	C105544	C_6_H_12_O_2_	116.2	800.4	290.03	1.21	13.06 ± 0.52	79.82 ± 7.13	12.84 ± 1.66	12.24 ± 1.95
Heptanal dimer	C111717	C_7_H_14_O	114.2	899.4	410.27	1.70	684.87 ± 46.75	285.09 ± 14.02	780.97 ± 35.21	395.66 ± 12.06
Heptanal monomer	C111717	C_7_H_14_O	114.2	899.3	409.93	1.33	281 ± 21.65	212.67 ± 8.64	296.56 ± 11.67	71.12 ± 2.49
Heptanoic acid dimer	C111148	C_7_H_14_O_2_	130.2	1053	693.30	1.76	91.78 ± 6.98	1362.95 ± 189.03	190.7 ± 22.46	81.34 ± 6.96
Heptanoic acid monomer	C111148	C_7_H_14_O_2_	130.2	1051.6	690.58	1.36	217.9 ± 9.68	457.72 ± 51.97	86.71 ± 5.29	60.26 ± 1.53
Hexanal dimer	C66251	C_6_H_12_O	100.2	791.1	279.29	1.57	1899.55 ± 72.68	2897.65 ± 52.78	2694.57 ± 86.19	1494.49 ± 19.81
Hexanal monomer	C66251	C_6_H_12_O	100.2	791.1	279.29	1.26	1088.22 ± 49.57	1063.24 ± 43.75	1230.44 ± 37.1	1166.24 ± 21.4
Isoamyl acetate dimer	C123922	C_7_H_14_O_2_	130.2	876	378.14	1.75	26.48 ± 6.98	144.48 ± 13.61	20.43 ± 1.35	20.8 ± 2.31
Isoamyl acetate monomer	C123922	C_7_H_14_O_2_	130.2	876.6	378.85	1.30	66.4 ± 4.29	395.1 ± 12.85	102.52 ± 8.51	54.28 ± 6.8
Linalool oxide dimer	C60047178	C_10_H_18_O_2_	170.3	1076.6	736.77	1.82	89.87 ± 6.51	450.59 ± 73.39	63.83 ± 5.24	49.68 ± 5.15
Linalool oxide monomer	C60047178	C_10_H_18_O_2_	170.3	1077	737.67	1.26	326.91 ± 24.45	804.93 ± 72.82	74.35 ± 25.8	34.87 ± 2.02
Methyl 2-methylbutanoate dimer	C868575	C_6_H_12_O_2_	116.2	768.8	258.26	1.54	217.3 ± 9.99	252.67 ± 7.51	24.62 ± 2.9	15.52 ± 1.03
Methyl 2-methylbutanoate monomer	C868575	C_6_H_12_O_2_	116.2	770	259.28	1.18	404.26 ± 20.26	352.79 ± 23.05	211.43 ± 8.69	146.93 ± 5.49
Pentanal dimer	C110623	C_5_H_10_O	86.1	694.6	195.77	1.43	1044.79 ± 43.21	923.43 ± 31.09	1084.99 ± 23.53	1093.19 ± 19.8
Pentanal monomer	C110623	C_5_H_10_O	86.1	694.9	196.05	1.18	964.72 ± 56.23	1044.44 ± 46.97	749.54 ± 55.68	321.03 ± 3.04
Propylacetate dimer	C109604	C_5_H_10_O_2_	102.1	707.7	206.83	1.48	39.05 ± 17.76	420.42 ± 8.88	27.48 ± 7.19	20.46 ± 0.21
Propylacetate monomer	C109604	C_5_H_10_O_2_	102.1	708.4	207.36	1.17	64.59 ± 6.93	292.21 ± 5.69	71.38 ± 3.49	83.29 ± 4.01
Propylsulfide	C111477	C_6_H_14_S	118.2	884.7	388.21	1.16	42.7 ± 2.2	39.03 ± 1.77	226.35 ± 16.02	80.77 ± 4.8

## Data Availability

Not applicable.

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
