# Peer review of "Characteristic Volatile Fingerprints of Four Chrysanthemum Teas Determined by HS-GC-IMS"

_molecules, 2021, doi:10.3390/molecules26237113_

Round 1

Reviewer 1 Report

The study was investegated the volatile profiles and contents of ‘Xiaokuixiang’ vis-à-vis ‘Hangju’, ‘Huangju’, and chamomile tea, by GC–IMS, and their flavor fingerprints were established. PCA was also used as an exploratory technique to discriminate the four chrysanthemum teas. The paper is well written and provided a nice approach and results. However, some information should be explained in order to understand and enhance the work quality.  

The authors stated that ''To classify chrysanthemum teas according to their characteristic volatile compounds, a PCA was implemented. PCA is a multivariate statistical method used to analyze the correlation between multiple variables''.  The PCA is an exploratory technique dedicated to data visualization, and not classification. The PCA is used  TO DETECT the existence of PATTERNS and INNER RELATIONSHIPS. So, the selection of words should be correlated with the purpose of the technique.  PCA does NOT classify Classification implies supervision or imposing limits, boundaries, or thresholds. However, one piece of advice also is to investigate the data to construct a classification model to classify the four teas, for example, PLS-DA or OPLS-DA could be a nice idea to enhance the quality of this paper. On the other hand, the results were visualized by using PCA score plots and loading plots. The PCA discussion is limited, and no information about the explained variances. Could you please ad the explained variances in the graph and in the discussion?  However, the scores should be linked and interpreted in relation to the loading plots. For example, the scores plots should be explained using the loading plots, from those loading graphs in the discussion indicate which chemicals are responsible for the discrimination, and each tea is characterized by which chemical (or group of chemicals). Please add and explain why? Please add in your discussion all those information.

Which kind of data preprocessing did you use for PCA? Please add and explain why? if any preprocessing was applied also please explain why? All those remarks should be explained within the discussion and detailed in order to understand the work. The new corrected draft should be concentrated on constructing a classification model with more details for PCA analysis. 

Figure 2. The three-dimensional spectrum of VOCs from four chrysanthemum teas. This figure is not clear please replace it with a 2D chromatogram profile (retention time vs intensity). On the other hand,  Supplementary Table S2 was not well explained. The discussion is limited, and no clear conclusion, please explain or delete it. 

Reviewer 2 Report

The authors report the characterization of teas based on four varieties of chrysanthemum (dry flowers). As the authors indicate, characterization studies of volatile compounds (in comparison to non-volatile ones) in these varieties of chrysanthemum by the method used by them (S – GC – IMS) are practically non-existent. However, the technique and type of grouping analysis analyzed by them is similar to that reported for other varieties of chrysanthemum (eg DOI: 10.4314 / tjpr.v15i10.24, 10.1093 / chromsci / 46.2.127, 10.1039 / C7RA13503C, 10.1080 / 22297928.2011 .10648201,10.1002 / ffj.1712). Minor changes could improve the scientific contribution of the study as follows: 

General

  • English grammar and syntax are good, yet the manuscript comprehension will improve even more if it is reviewed by a native-English spoken colleague or translation agency.
  • In certain cases, supplementary tables/figures contain more important information than those provided in body text (see specific comments below).

Specific

  • Title. Suggestion: “…and PCA of their volatile compounds” could be eliminated.     
  • Abstract. It should be more concise without sacrificing important results, expressed in a more quantitative way, including statistical differences (p-values). For example, according to Table S1, 47 volatiles of diverse chemical nature were identified, yet only 10 of them (1,8-cineole dimer / monomer, 1-menthol monomer, 2-methylbutanal dimer, 6-methyl-4-hepten-2-one, ethyl-2-methylbutanoate, hexanal & pentanal monomer / dimer) account for >50% of the relative abundance (Figure 5) of variety-specific form [Huan (~ 43, 892)> Xiao (~ 24, 307 )> Cham (~ 21, 076)> Hang (~ 16, 067)].
  • Introduction. This section should be shorter and argumentative sentences/paragraphs should be placed in the discussion section. Figure 1 should be included in this section [line 93: “…chamomile tea (Figure 1)]
  • Figures. Their resolution should be improved (≥300 dpi, particularly figure 6) and the B&W color scale could be better. Figure 1 could be firstly mentioned (and located) in the introduction, Figures 2-4 and 6 could be included as supplementary material since they do not add further information as to that provided by the remaining tables and figures, otherwise explain them in-depth. Include information as to PC1/PC2/PC3´s explained variance and include eigenvectors for chemical groups to rapidly identify which group of volatiles (those indicated in figure 5) explain the most inter-sample differences. Figure 7 should be sharper (higher size) while Figure 8 could be downsized.
  • Tables. Statistical differences between samples should be indicated when needed an please include as footnotes any complementary information (e.g. abbreviations, statistical significance, etc). Table 2 (Line 168) could not be identified, Table S1 should be included within the body text since very relevant information can be identified from it (e.g. Hangju tea is the best source of menthol monomer). Table S2 is very crowded, to reduce its size, the legend "Values (distances?) are expressed as x 107" could be included as a footnote to the table and leave only values with one or two decimal places.
  • Results and discussion. Be consistent with all abbreviations throughout the manuscript, including their meanings the first time they are mentioned. Even though the discussion is well supported with descriptive data (binary data, figures), the authors should intent to give a deeper explanation as to the associated factors related to inter-sample variations, in a more comparative way with preceding studies on the subject. Authors should put special attention to subtle differences in important health-promoting and odor-promoting compounds, regardless of their variety-specific content.

Round 2

Reviewer 1 Report

The paper was corrected as suggested. The paper quality is enhanced.